# Host Components That Modulate the Disease Caused by hMPV

**DOI:** 10.3390/v13030519

**Published:** 2021-03-22

**Authors:** Nicolás M. S. Gálvez, Catalina A. Andrade, Gaspar A. Pacheco, Jorge A. Soto, Vicente Stranger, Thomas Rivera, Abel E. Vásquez, Alexis M. Kalergis

**Affiliations:** 1Departamento de Genética Molecular y Microbiología, Facultad de Ciencias Biológicas, Millennium Institute of Immunology and Immunotherapy, Pontificia Universidad Católica de Chile, Santiago 8320000, Chile; nrgalvez@uc.cl (N.M.S.G.); cnandrade@uc.cl (C.A.A.); grpacheco@uc.cl (G.A.P.); jasoto6@uc.cl (J.A.S.); vstranger@uc.cl (V.S.); tlrivera@uc.cl (T.R.); 2Sección de Biotecnología, Agencia Nacional de Dispositivos Médicos, Innovación y Desarrollo, Instituto de Salud Pública de Chile, Ñuñoa 7750000, Chile; avasquez@ispch.cl; 3Facultad de Medicina y Ciencia, Universidad San Sebastián, Providencia, Santiago 7500000, Chile; 4Departamento de Endocrinología, Facultad de Medicina, Pontificia Universidad Católica de Chile, Santiago 8320000, Chile

**Keywords:** human metapneumovirus, innate immunity, adaptive immunity, host factors

## Abstract

Human metapneumovirus (hMPV) is one of the main pathogens responsible for acute respiratory infections in children up to 5 years of age, contributing substantially to health burden. The worldwide economic and social impact of this virus is significant and must be addressed. The structural components of hMPV (either proteins or genetic material) can be detected by several receptors expressed by host cells through the engagement of pattern recognition receptors. The recognition of the structural components of hMPV can promote the signaling of the immune response to clear the infection, leading to the activation of several pathways, such as those related to the interferon response. Even so, several intrinsic factors are capable of modulating the immune response or directly inhibiting the replication of hMPV. This article will discuss the current knowledge regarding the innate and adaptive immune response during hMPV infections. Accordingly, the host intrinsic components capable of modulating the immune response and the elements capable of restricting viral replication during hMPV infections will be examined.

## 1. Introduction: Human Metapneumovirus

### 1.1. The Disease Caused by hMPV

Human metapneumovirus (hMPV) is a respiratory virus from the *Pneumoviridae* family, first described in 2001, when it was isolated from the respiratory tract of children from the Netherlands [1]. hMPV represents one of the leading causes of acute respiratory tract infections (ARTI) in children, immunosuppressed patients, and the elderly [2,3]. This pathogen is also considered a primary cause of death in infants under five years old [4,5]. Several studies have shown that hMPV is highly prevalent worldwide, affecting up to 86% of the global population of infants under five years old [6,7,8,9]. This virus represents a significant economic burden on health care systems worldwide [6]. In the USA alone, approximately 20,000 hospitalizations are registered every year due to this virus, with a cost per infected patient ranging between USD 3850 and USD 9946 [6].

Clinical signs and symptoms associated with hMPV are mainly respiratory problems ranging from coughing, wheezing, and fever to more severe complications, such as bronchiolitis and pneumonia [1,10]. hMPV mainly infects and affects the lower respiratory tract (LRT), with the requirement for mechanical ventilation in the most severe cases [1,10]. Infection with hMPV has also been associated with the manifestation of neural-related symptoms, such as encephalitis and febrile seizures [11,12,13]. The most severe symptoms associated with hMPV infections are often reported in infants between younger than one year old, but are highly prevalent during early childhood up to five years old [1,10]. High-risk factors in infants include asthma, preterm birth, and previous infections with other respiratory viruses, such as the human respiratory syncytial virus (hRSV, recently renamed human orthopneumovirus), and these predispose infants to a more severe disease manifestation after an hMPV infection [14]. Regarding its epidemiology, hMPV begins circulating among the general population during winter and lasts until the end of spring [15].

Because hMPV is a relevant respiratory virus, the need to understand how the immune system contributes to controlling this infectious agent can help develop new vaccines and therapies [16,17]. This article will discuss the current data relative to the host immune system’s (both innate and adaptive) capability to detect the various components of hMPV. Further, intrinsic components and restriction factors belonging to the host, which modulate the immune response against this pathogen, will also be discussed.

### 1.2. Proteins, RNA, Viral Structure, and Infective Cycle

hMPV is a virus with a single-stranded (ss), negative-sense, and non-segmented RNA genome that is approximately 13.3 kb in size [18,19]. This virus encodes nine structural proteins, each with different functions. The order in which the genes for these proteins can be found in the viral genome is the following: 3′-N-P-M-F-M2(-1/-2)-SH-G-L-5′ [18]. The nucleoprotein (N—43.5 kDa) is responsible for the encapsidation and the protection of the genomic ssRNA, binding to it [20]. The phosphoprotein (P—32.4 kDa) is a co-factor of the L protein, required for stabilization and the synthesis of new genetic material upon interaction with the RNA-N protein complex [21]. The inclusion bodies commonly detected during hMPV infections are mostly formed by these two proteins, N and P [22]. The matrix (M—27.6 kDa) protein aids in viral assembly and budding, possesses a high-affinity binding site for Ca2^+^, and is also the major component of the virus [23]. Interestingly, this protein seems to be secreted in a soluble form by infected cells and can induce the secretion of inflammatory cytokines in in vitro cultures [24]. The fusion protein (F—58.4 kDa) is responsible for the virus-cell binding and membrane fusion [25]. The M2-1/2 proteins (21.2 and 8.1 kDa) play a role in modulating the processivity of the RNA polymerase and are also responsible for the modulation of the immune response elicited by the host [18]. The small hydrophobic protein (SH—20.9 kDa) plays different roles in the modulation of the innate immune response (such as the inhibition of the interferon (IFN) response) and may also have a function as a viroporin [26]. The glycoprotein (G—25.7 kDa) binds to cellular glycosaminoglycans and is responsible for the attachment of the viral particles to the cells of the host [27]. Moreover, there is compelling evidence indicating that the G protein contributes to the inhibition of the IFN-I response [28], as well as contributing to the recruitment of neutrophils in the airways through enhanced secretion of the chemoattractants CXCL2, CCL3, CCL4, IL-17, and TNF [29]. Lastly, the large polymerase protein (L—230.6 kDa) has binding sites for zinc, with a multifunctional catalytic activity, and is the one responsible for the synthesis of new genetic material, along with the cofactors described above (Figure 1A) [1,18,30,31]. It is important to emphasize that the gene encoding the M2 protein contains two open reading frames (ORF) that lead to the expression of either the M2-1 or M2-2 protein [18]. The viral particles from hMPV exhibit a pleomorphic morphology, ranging from spherical to filamentous forms, and a lipid envelope with protein projections of about 13–17 nm [1,31]. The airway epithelial cells (AECs) are the primary infection target of hMPV, to which the virus attaches itself due to the interaction between the G protein and heparin sulfate expressed on the surface of AECs [30]. Cell surface attachment is followed by the engagement of the F protein to an integrin located on the surface of AECs, promoting the viral fusion to the cell membrane and the entry of the genetic material of hMPV (and some of its proteins) into the host cell [32]. Once inside, the RNA polymerase transcribes the viral negative-sense RNA into a monocistronic positive-sense mRNA, allowing for the translation of this new RNA molecule into viral proteins (Figure 1B) [30]. The viral glycoproteins can be transported via the Golgi apparatus to the membrane and accumulate for the assembly of new viral particles [30]. When the production of viral proteins reaches a threshold concentration, the RNA polymerase replicates the genome into positive-sense RNA. This positive-sense RNA will be used as the template for the new genomic negative-sense RNA, which will then be contained within the new viral particles [30]. This replication process takes place within cytoplasmic inclusion bodies, created by the interaction between the N and P proteins [22]. Finally, with the support of the M protein, the viral particles can be assembled and released from the cell surface by budding from the membrane [30]. The morphogenesis of filamentous viral particles has been shown to colocalize with F-actin and lipid raft microdomains, and both of these interactions seem to be mediated by specific domains of the G protein [33]. It has been suggested that the SH protein can play an important role in this reproduction cycle through the permeability of the membrane; however, this needs to be further studied [26].

Several of these proteins have been chosen as targets for the development of antiviral treatments against hMPV, given that their roles in the infective cycle have been well described [16,17,34,35,36,37,38]. However, many features of this cycle still remain unclear. Recent reports using a three-dimensional model of human airway epithelial (HAE) tissue studied the interaction of the anti-hMPV 54G10 neutralizing antibody over the infective and spreading capabilities of this virus [39]. Recent results have suggested that some of the spreading of hMPV in HAE tissue may require a cell-to-cell mechanism [39]. Furthermore, some studies have shown that an hMPV infection induces the reorganization of the actin cytoskeleton, allowing this direct cell-to-cell spread [31]. Remarkably, inhibition of the actin polymerization reduced the spreading of hMPV [31]. Taken together, these results show that hMPV has other strategies to infect nearby cells that may not require exclusively the F protein. However, further studies are needed to better understand the role of other viral proteins in this process, and this may open new possibilities in the development of effective treatments.

The replication cycle of hMPV has been a subject of study, and the contribution of the viral proteins to this process can be significantly relevant, but the persistence of the virus in the lungs is also of interest. Not long ago, it was thought that an infection with hMPV could be cleared 7 days after the initial exposure [40]. However, some studies have provided evidence of biphasic growth kinetics of hMPV [40]. Infection of BALB/c mice with the hMPV/CAN98-75 strain showed that this virus exhibited two viral titer peaks, one at 7 days post-infection (dpi) and the other at 14 dpi, with matching peaks of weight loss on those days [40]. This was the first report that suggested the persistence of hMPV in the lungs [40]. However, studies performed in cotton rats showed only single-phasic growth kinetics [41,42,43], as was also observed in several other studies where the hMPV/D03-574 strain was used to infect BALB/c mice [44]. These differences could be explained by the different strains of hMPV used in each study [40,41,42,43,44]. Along these lines, hMPV has one serotype that can be separated into two main groups depending on their genotype (A and B) [30], which can be further divided into four subgroups (A1, A2, B1, and B2) [45], with the A2 subgroup divided into A2a, A2b, and A2c, and the B2 subgroup into B2a and B2b [6]. These genotypes of hMPV can circulate simultaneously during the year, and the dominant subgroup often changes each year [6]. Interestingly, it was shown that the genotype B was associated with more disease signs [46].

## 2. Innate Immune Response and Components Recognizing hMPV

The response elicited by the innate immune system upon hMPV infection plays an essential role in the primary control of the disease and the subsequent activation of the adaptive immune response [47,48,49,50,51,52]. In the following section, the innate immune response elicited during infections with hMPV will be discussed as well as how the innate immune cells activate the interferon pathways. Additionally, how the innate and intrinsic host components can modulate the disease caused by hMPV will be discussed.

### 2.1. Cells of the Innate Repertoire and the IFN Pathway

IFNs are perhaps the most extensively studied antiviral factors secreted in response to the detection of viral infections [53,54]. These cytokines can activate transcriptional programs in infected and noninfected cells, ultimately restricting viral replication [53,54]. Therefore, a thorough revision of antiviral factors must consider the role of IFNs as part of the extensive antiviral network that the immune system employs to control viral infections. The innate immune system is responsible for the rapid initial control of infections and initiating the adaptive immune response [53,54]. Thus, in this section, our focus will be on the innate immune cells, particularly on dendritic cells (DCs) and alveolar macrophages (AMs), both relevant for the control of infections caused by hMPV [49].

As mentioned above, IFNs constitute one of the principal mechanisms for controlling viral replication [53]. IFNs are small proteins secreted by most cells upon infection that alert neighboring cells and promote an antiviral state [54]. There are three major classes of IFNs (type I, type II, and type III) [54,55,56,57,58,59], each with variable specificity for the various IFN receptors that cells express [54]. Different cell types secrete different IFNs, and various cell types also express their respective receptors. While the IFN-I receptor complex (IFNAR) is present in every nucleated cell type, the IFN-II and IFN-III receptor complexes (IFNLR and IFNGR) are present almost exclusively in epithelial cells and immune cells, respectively [60,61]. During an hMPV infection, engagement of IFNs and their receptors seems critical for controlling viral spreading in the host, as IFNAR-deficient mice exhibit higher viral loads in the lungs and an inadequate virus-specific cellular immune response [48]. The signaling pathways elicited by IFNs involve different IFN receptors that may induce the phosphorylation of STAT transcription factors by JAK or TYK protein kinases, which are in turn associated with receptors with tyrosine kinase activity [53,54]. Phosphorylated STAT dimers can promote the transcription of interferon-stimulated genes (ISGs) upon binding to specific sequences of DNA [62]. IFN-I and IFN-III signaling pathways converge in the phosphorylation of a STAT1/2 heterodimer [62]. Thus, specific ISGs stimulated by either IFN-I or IFN-III are often similar [60]. On the other hand, the binding of IFN-γ to its receptor induces the phosphorylation of a STAT1 homodimer and the subsequent transcription of ISGs that possess gamma-activated sequences (GAS) in their promoters, which are often associated with genes encoding for proteins that participate in inflammatory responses [62].

ISGs are diverse and often constitute a mixture of antiviral effector proteins and transcription factors that further promote the expression of more antiviral effectors or other regulatory proteins [53,63]. Moreover, some ISGs are viral sensors, often overexpressed to more efficiently detect infecting viruses [53], while other ISGs encode proteins that can help desensitize the IFN activation pathway [64]. The most evident outcome of the activation of the IFN pathway is the synthesis of antiviral effector proteins. These proteins can help prevent infections by disrupting the viral replication cycle, not only in target cells such as epithelial cells, but also in immune cells. Some well-known effector proteins are those of the tripartite-motif-containing (TRIM) family, the interferon-induced transmembrane (IFITM) protein family, viperin, RNaseL, and CH25H, among several others [53,65,66]. These proteins can act at any given point of the viral replication cycle: fusion, replication, transcription, translation, assembly, and budding. Although there is little information regarding which of these effector proteins are responsible for hMPV restriction, studies have shown that the TRIM and IFITM families could work as significant drivers of viral restriction.

The interferon response, particularly that of the respiratory epithelium, is so critical for hMPV restriction that the virus has acquired ways to halt IFN-I secretion [47]. In this line, the SH protein of hMPV is capable of inhibiting STAT1 phosphorylation (Table 1) and thus inhibiting the transcription of many ISGs [67,68]. Some hMPV strains possess a P protein capable of inhibiting the sensing of 5′-triphosphate viral RNA by RIG-I (Table 1) and therefore inhibit the production of IFN-I and the subsequent expression of ISGs [69]. Interestingly, it has been shown that genotypes A and B of hMPV elicit the secretion of IFN-III in nasopharyngeal aspirates from infected children, while IFN-β secretion is elicited only by genotype A [70]. Moreover, IFN-γ appears to be virtually absent in aspirates from hMPV-infected children, or at least, a limited amount is secreted in these infected children [71,72].

**Table 1 viruses-13-00519-t001:** Described role of the hMPV proteins on the host components of the immune system.

hMPV Proteins	Impact on the Host Immune Response	References
**Nucleoprotein**	An epitope from this protein promotes a protective CTL response. Along with the P protein, this is the other main component of the inclusion bodies reported during hMPV infections	[20,22,73]
**Phosphoprotein**	Restricts the ability of RIG-I to recognize 5′-triphosphate viral RNA, weakening the expression of IFN-I and ISGs	[21,69]
**Matrix protein**	It is secreted by infected cells in a soluble form and induces the secretion of inflammatory cytokines	[23,24]
**M2-1 protein**	An epitope from this protein stimulates a protective CTL response	[18,73]
**M2-2 protein**	Prevents the homodimerization process of IRF7, resulting in the lack of IFN-α induction from the TLR7 signaling pathway Forms a complex with MyD88 and inhibits TLR-driven signaling	[18,74,75]
**Small hydrophobic protein**	Blocks the phosphorylation process of STAT1, reducing the transcription levels of ISGs Inhibits the TLR7 signaling pathway, decreasing IFN expression It might be involved in decreasing the activation of CD4^+^ T cells	[67,68,76,77]
**Glycoprotein**	Might participate in reducing the activation of CD4^+^ T cells Forms a complex with RIG-I to avoid viral sensing Contributes to neutrophil recruitment via enhanced secretion of CXCL2, CCL3, CCL4, IL-17, and TNF	[27,28,29,77]

CTL: Cytotoxic T cells; P: Phosphoprotein; RIG-I: Retinoic acid-inducible gene I; IFN-I: Type I Interferon; ISGs: Interferon Stimulated Genes; IRF: Interferon Regulatory Factor; TLR: Toll-like receptor; MyD88: Myeloid-Differentiation Factor 88; STAT1: Signal transducer and activator of transcription 1.

Although it is widely accepted that the IFN response is elicited as a consequence of hMPV replication and that IFNs are also capable of restricting viral replication [47], there is a general lack of information about which factors are responsible for said restriction. Members of the IFITMs family, such as IFITM1, IFITM2, and IFITM3, can inhibit the fusion of the viral envelope and the cell membrane during hMPV infections [78,79,80]. These are other genes reported to exhibit antiviral capacities during hMPV infections: heparanase (HPSE), which cleaves heparan sulfate and could reduce the binding of the G protein to the target cell [80,81]; CD9, a surface protein expressed by epithelial cells and B cells, implied in tyrosine kinase signal transduction [80,82]; and P2RY6, a pyrimidinergic receptor responsive to UDP, UTP, and ADP, which mediates inflammatory responses, possibly via the sensing of pyrimidine nucleotides released into the extracellular fluid after cell lysis [80].

DCs and AMs represent major immune sources of IFN-I and IFN-II during infections with respiratory viruses [83,84]. These cells are among the relatively few cell types capable of secreting IFN-γ, which is critical for promoting antiviral immunity [83,84]. These cell types are responsible for antigen presentation and possess a diverse collection of molecules contributing to viral recognition [83,84]. The engagement of these receptors usually leads to the transcription of antiviral and proinflammatory genes, further discussed in the following sections. Plasmacytoid DCs (pDCs), a specialized subset of DCs, are responsible for much of the secretion of IFN-I in response to viral infections [83,84]. In particular, pDCs produce IFN-α, which gives them an essential role in establishing an antiviral immune response [83,84]. These innate immune cells promote an antiviral state in target cells, such as epithelial cells. This antiviral state is achieved by secretion of IFN-I and tissue inflammation induction, which recruits and activates other effector immune cells [83,84]. Interestingly, pDCs have shown low susceptibility to being infected with hMPV, and even when infected with a strain of hMPV that can avoid detection by RIG-I, they secrete large quantities of IFN-I [69].

### 2.2. Intrinsic Host Components and Factors Recognizing the Genetic Material of hMPV

Viral infection with hMPV leads to the recognition of components derived from this virus, which initiates the secretion of IFNs [85,86,87]. Viral components are recognized by pattern recognition receptors (PRRs), which are responsible for the identification of pathogen-associated molecular patterns (PAMPs) [85,86,87]. Each PRR can recognize different types of PAMPs, such as nucleic acids or proteins from hMPV, and are found in various locations, including cellular and endosomal membranes, the cytosol, and on the surface of cells found in the bloodstream or interstitial fluids [85,86]. The various PRRs can be classified into Toll-like receptors (TLRs), retinoic acid-inducible gene I (RIG-I)-like receptors (RLRs), NOD-like receptors (NLR), and C-type lectin receptors (CLRs), among others [85,87].

The receptors that can recognize the genetic material from viruses comprise about ten different PRRs in humans [86,88]. More specifically, TLR3, TLR7, and TLR8 can recognize the RNA from hMPV, and they will be further characterized below [74,76,86,88,89,90,91,92,93,94,95].

TLR3 can recognize viral dsRNA and is expressed by several cell types, including conventional dendritic cells (cDCs), macrophages, epithelial cells, fibroblasts, and cells from the central nervous system [96]. Depending on the cell type, TLR3 can be located both on the surface of the cell and inside endosomes, as is the case for fibroblasts, or only within endosomes, as is the case for cDCs [96]. Once TLR3 binds to dsRNA, the adaptor molecule TIR-domain-containing adapter-inducing interferon-β (TRIF) becomes activated and initiates the phosphorylation and nuclear translocation of the transcription factors nuclear factor κB (NF-κB) and interferon regulatory factor 3 (IRF3) (Figure 2B) [97,98,99]. While NF-κB induces proinflammatory cytokines, IRF3 promotes the expression of IFN-I, such as IFN-β [97]. Notably, the interaction between TLR3 and hMPV has not been studied thoroughly yet. In vitro studies using A549 cells have demonstrated that the expression of TLR3 is upregulated upon infection with hMPV, with a peak at 9 hours post-infection (p.i.) [90]. Consistently, in vitro studies with monocyte-derived DCs (moDCs) showed that infection with hMPV leads to an increased expression of TLR3 [91]. Furthermore, hMPV-infected BALB/c mice reported an increased expression of TLR3 in the lungs, reaching its peak at 5 days p.i. [90]. Interestingly, when BALB/c mice were inoculated with both hMPV and polyinosinic-polycytidylic acid (polyI:C), a synthetic analog of dsRNA, the expression of TLR3 was significantly increased [92]. Additionally, administration of polyI:C restricted the replication of hMPV and decreased lung inflammation in infected mice, suggesting that the restriction was a consequence of the rapid activation of TLR3 [92]. Additional studies relative to the interaction between TLR3 and hMPV are needed to soundly understand the contribution of this receptor to the immune response elicited during the disease caused by this virus. Some experiments that could be useful to address the role of TLR3 during hMPV infections may involve the use of mice knockout (KO) for or overexpressing TLR3 or the direct use of antagonists against TLR3 on in vivo assays, such as blocking antibodies. We would expect to see an increase in the viral clearance in the mice model overexpressing TLR3 mice, as the studies using polyI:C showed a protective role for this PRR. On the contrary, we would expect to see a decrease in the viral clearance using the TLR3-KO or TLR3-blocked mice, again in line with the protective role described for this receptor. Finally, the measurement of different genes related to this receptor and its signaling pathways, through transcriptomic analyses, will help to understand the relationship between TLR3 and hMPV.

TLR7 and TLR8 can recognize ssRNA, and they can be found in various cell types, such as neurons and immune cells [93,94]. However, these receptors are differentially expressed by various immune cells [93,94]. In this line, TLR7 is more expressed in pDCs, monocytes, and B cells, while TLR8 is mainly expressed by monocytes, macrophages, cDCs, and neutrophils [93,95]. Both of these TLRs are intracellular receptors located within endosomes, and they promote the secretion of proinflammatory cytokines and IFN-I [95]. TLR7 activation promotes the secretion of IFN-β and the phosphorylation of NF-κB at early time points during viral infections, which subsequently decrease as the secretion of IFN-α increases (Figure 2B). The activation of TLR8 only promotes the secretion of INF-β and the phosphorylation of NF-κB during the early stages of viral infections, with a later reduction in the respective secretion and phosphorylation of these molecules [95]. Particularly, the RNA from the subtypes A1 and B1 of hMPV can be detected by TLR7 in pDCs [69]. An in vitro infection with hMPV leads to increased expression of TLR7 and TLR8, with peaks at 12 hours p.i., followed by a decreased expression after this time point [90]. The expression of TLR7 and TLR8 in the lungs of BALB/c mice was upregulated during infections with hMPV, reaching peaks by day 5 p.i. [90]. During in vitro and in vivo experiments, the expression of TLR7 and TLR8 was significantly higher than that of TLR3 [74]. This increased expression suggests that TLR7 and TLR8 may be more relevant than TLR3 in modulating infections with hMPV [74]. Interestingly, the M2-2 protein of hMPV inhibits the homodimerization of IRF7 (Table 1), which results in the inhibition of the induction of IFN-α that depends on the TLR7 signaling pathways [74]. The M2-2 protein has also been shown to inhibit MyD88 signaling during TLR-driven responses in monocyte-derived DCs (moDCs), suggesting that this is one of the main molecular sensors targeted by hMPV to avoid detection and further diminish IFN-I secretion by DCs [75]. Even more, the SH protein of hMPV can block the signaling pathway induced by TLR7 and MyD88 in pDCs (Table 1), preventing the expression of IFN-I and favoring viral replication [76].The secretion of IFN-I induced by the activation of TLR7 and TLR8 can be prevented by activating proteins of the suppressor of the cytokine signaling (SOCS) family [89]. Viruses commonly modulate SOCS proteins in order to evade the innate immune response [89]. The activation of TLR7 leads to the expression of SOCS1 and SOCS3 proteins, and through a negative feedback loop, these proteins impair the secretion of IFN-I mediated by TLR7 signaling [89].

Receptors expressed on the membrane or within endosomes of the cells that can recognize viral RNA from hMPV have been addressed above. Nonetheless, viral RNA can also be found in the cytosol of the cell [86]. RLRs are the receptors in charge of recognizing and inducing a response against RNA molecules found in the cytosol of the cells [86]. Among the different RLRs are RIG-I (which recognizes short dsRNA and 5′-triphosphate RNA) and melanoma differentiation-associated gene 5 (MDA5) (which recognizes long dsRNA) [86]. RIG-I and MDA5 can be found in fibroblasts, epithelial cells, and immune cells, and the detection of viral RNA by either of these receptors leads to the secretion of IFN-I and the expression of ISG (Figure 2B) [100]. In vitro experiments with A549 cells showed that hMPV could promote the expression of RIG-I [101]. Simultaneously, downregulation of this molecule decreases IFN-β and IL-8, increasing the replication of hMPV [101]. Additionally, the protein mitochondrial antiviral signaling (MAVS), an adaptor for RIG-I, is vital for the induction of this signaling pathway [101]. Both A1 and A2 subgroups from hMPV can be detected by RIG-I, inducing the expression of IFN-α/β and IFN-β, respectively [69]. Sensing of viral RNA by RIG-1 induces the expression of carcinoembryomic antigen-related cell adhesion molecule 1 (CEACAM1), which is a tyrosine-based inhibitory motif (ITIMs) immunoreceptor [102]. The expression of this immunoreceptor leads to an inhibition in the replication of hMPV, through limiting the production of its proteins [102]. Apparently, detection of viral genetic material initiated by RIG-I is critical to defend against hMPV infections, since at least two viral proteins are capable of impairing RIG-I-mediated viral RNA detection [28]. The G protein has been shown to form a complex with RIG-I, which could explain the diminished phosphorylation and/or nuclear translocation of NF-κB and IRF proteins, as well as the impaired IFN-I, cytokine, and chemokine responses, attributed to the presence of the G protein in infecting viral particles [28]. On the other hand, the P protein from the B1 subgroup has been shown to impair the detection of the viral RNA from hMPV through RIG-I (Table 1) [69].

Even though hMPV can induce the expression of MDA5 in A549 cells, this protein does not play an essential role in the induction of an “alert” response against this virus [101]. However, in vitro experiments using moDCs demonstrated that MDA5 plays a crucial role in the induction of the IFN-I responses against hMPV [103]. In vivo experiments have supported this finding and further demonstrated that MDA5 impairs the viral clearance and enhances the inflammation in the lungs, promoting more severe diseases during hMPV infections [103].

Another type of receptors located in the cytosol of the cells is the NLRs [86]. NOD2 belongs to this receptor type, and it recognizes ssRNA from viruses, promoting an antiviral response [104]. Even though NOD2 has been shown to activate IRF3 and induce the secretion of IFN-I during *Pneumoviridae* infections [104], no studies addressing the role of NOD2 during hMPV infections have been issued to date. It would be interesting to evaluate whether hMPV can activate NOD2 and therefore promote its signaling through similar pathways to other *Pneumoviridae* viruses.

Several intrinsic antiviral factors can prevent viral replication within the host cells [78,105,106]. Among these restriction factors are the IFITM proteins, which can block the entry of the virus into the cell (preventing the fusion of the virus with the membrane) and promote the secretion of IFNs (Figure 2A) [105,106]. Within this family, the IFITM3 protein prevents the infection of subgroups A1 and B1 of hMPV in vitro [78]. The IFITM3 protein is the first restriction factor to be described that can prevent the infection by hMPV.

TRIM proteins have also been implicated in the immune response against viruses, whether as restriction factors or by modulating the immune response [107]. Playing a role in the immune response are several TRIM proteins from which TRIM56 has been described as an intrinsic antiviral factor for RNA viruses [107,108]. TRIM56 prevents the replication of some viruses and promotes the secretion of IFNs through the activation of TLR3 [107,108]. Even so, it has been suggested that TRIM56 can directly detect viral RNA [109]. However, TRIM56 was unsuccessful in inhibiting the replication of hMPV on its own [108]. Nonetheless, it would be interesting to evaluate whether other TRIM proteins can affect the replication of hMPV.

### 2.3. Intrinsic Host Components and Factors Recognizing the Proteins of hMPV

The host cells have different mechanisms for regulating and controlling the damage induced during pathogenic infections. Notably, hMPV has specialized evasion mechanisms that suppress or polarize the immune response toward differentiated profiles not suitable for carrying out viral clearance [49]. This section will address the molecules that the host employs to modulate the viral replication and the pathology induced during an hMPV infection.

Protease-activated receptor 1 (PAR1) is a member of the G-protein-coupled receptors (GPCRs) [110]. PAR1 has been associated with the coagulation process [110] and other physiological activities [111,112]. A role for PAR1 has also been reported during viral infections, such as those promoted by influenza virus, herpes simplex virus, hMPV, and even dengue virus [113,114,115,116]. To date, the specific role of PAR1 remains controversial [113,117]. PAR1 contributes to the inflammatory response during influenza virus infections and impairs viral replication, promoting a protective immune response [113,117]. It is unclear whether this dual response could be observed during infections with other viruses. Particularly for hMPV infections, the role of PAR1 has been evaluated in in vitro and in vivo assays [118]. In vitro experiments showed that PAR1 activation (which was achieved by the administration of its agonist, TFLLR-NH2) had no impact on the replication of hMPV [118]. However, the administration of an antagonist of PAR1, RWJ-56110, resulted in a dose-dependent decrease in viral titers [118]. In mice, the administration of the antagonist RWJ-56110, along with a lethal dose infection of hMPV (LD50), resulted in a decrease in weight loss (which correlated with a higher survival rate), as compared to untreated hMPV-infected mice [118]. Therefore, the inhibition or blocking of PAR1 might have beneficial results and decrease the exacerbated inflammatory immune response triggered during hMPV infections in mice (Figure 2A) [118].

MDA5 is essential for establishing an antiviral immune response through the secretion of IFNs [119]. The silencing of MDA5 from human moDCs infected with hMPV strongly decreased IFN-I and IFN-III expression [119]. However, IFN-II secretion was mostly unaffected [119]. When the role of MDA5 was evaluated in mice, similar results were obtained, confirming that MDA5 is essential for the activation of the IFN pathway against viral infections [103]. Interestingly, the lack of MDA5 in C57BL/6 mice (MDA5^−/−^ mice) affected the activation of the IFNs signaling pathways and generated a dysregulation of other proinflammatory cytokines, promoting an increase in lung pathology in mice [103]. Similar results have been reported for other respiratory viruses, such as rhinovirus [120] and murine norovirus [121], which suggests that MDA5 might be a master regulator of the antiviral immune response.

IRF3, IRF7, and IFN-β promoter stimulator 1 (IPS-1) are three proteins that have been involved in the activation of the IFN antiviral response [122]. The effects of these three proteins in the IFN immune response modulation during infection with hMPV are diverse [122,123]. The absence of both IRF3 and IRF7 promoted a synergic effect in the impairment of the antiviral response [123]. This impairment resulted in a decreased IFN response, increased lung damage, and increased viral loads, as seen in neonatal mice infected with hMPV [123]. However, the phenotype described was different when the host missed only one of these IRF molecules [123]. The absence of IRF3 resulted in a considerably decreased expression of IFN-β, IFN-II, and IFN-III, related to an increase in the viral load [123]. The absence of IRF7 induced a decrease in the expression of IFN-α4, IFN-β, IFN-II, and IFN-III; this effect, however, had no impact on the viral load. This result suggests that the depletion or blockage of IRF3 would be more relevant for the antiviral response than the blockage of IRF7 [123]. The depletion of IPS-1 was associated with a decrease in the expression of IFN-β and IFN-III [123]. Interestingly, upregulation of IFN-α was detected upon depletion of IPS-1, along with increased viral loads of hMPV and an exacerbated inflammatory response [123].

Recent reports have focused on understanding the mechanisms that different human cell types have for facing hMPV infections. Remarkably, the antiviral immune response associated with the IFN signaling pathways is dependent on the cell type (epithelial or immune) that elicits it [49,50]. The effect of hMPV infections in alveolar epithelial cells (A549), nasal epithelial cells (NEC), monocyte-derived macrophages (MDM), and monocyte-derived dendritic cells (MDDC) has been studied [49,50]. A differential expression profile of IFN molecules and IRFs was correlated with the different cells studied [49,50]. Two IFN molecules were mainly characterized: IFN-β was poorly expressed by epithelial cells, while IFN-I was expressed more strongly in MDM, MDDC, and epithelial cells. Within the IRFs evaluated, the expression of IRF1 was reported mainly in both MDM and MDDC, correlating directly with the expression of IFN-β [49,50]. In contrast, IRF7 expression was higher in A549 cells, MDM, and MDDC, correlating with the expression of IFN-I [49,50].

## 3. Components and Cells of the Adaptive Immune System Responding to hMPV

The antiviral response elicited by the cells from the adaptive branch of the immune system during infections with hMPV is broad and specific. This response is highly dependent on the interaction with the cells from the innate repertoire. In the following section, the adaptive immune response and the intrinsic host components related to hMPV and reported to date in lymphocytes (either T or B cells) will be discussed.

T cells express a vast array of genes that help modulate their activation and response [124]. These genes encode different types of proteins, such as surface receptors and cytokines [125]. Among the many receptors expressed by T lymphocytes, the T-cell receptor (TCR) is among the most characterized [124,126]. The TCR expressed by T cells is the result of a complex rearrangement of several genetic segments, which may lead to the expression of up to 10^20^ different individual TCRs [124,126,127]. Each of these TCRs can recognize different epitopes, giving T cells the capability to identify virtually any antigen [124,126]. The TCR expressed by CD4^+^ and CD8^+^ T cells must interact with the MHC-II and MHC-I receptors (respectively) expressed on the surfaces of different cells [124,126]. It is unclear whether hMPV can modulate the expression or the signaling pathways associated with the TCR. However, and as seen for other viruses, this is most likely possible [128]. Therefore, future studies should focus on addressing the capacities of hMPV to modulate TCR signaling and surface expression. A possible way to assess these modulating capacities could consider the characterization of the proteins previously described to interfere with the immunological synapse process in other *Pneumoviruses*. Because hMPV has fewer proteins than other *Pneumoviridae* viruses (i.e., it lacks NS1 and NS2), it is possible that most of its proteins acquired new functions to aid in the evasion of the immune response. Particularly, to evaluate if any protein is involved in the interference of the pMHC-TCR complex assembly, studies using artificial bilayers with purified T cells or cocultures between DCs and T cells in the presence of the virus or the proteins could be considered. The use of top-of-the-line microscopy techniques could also give insights into this issue.

The role of CD4^+^ T cells during hMPV infections has been poorly addressed to date. Nonetheless, it has been shown that hMPV is capable of inhibiting the activation of CD4^+^ T cells in vitro (Figure 2C) [51]. Upon infection of DCs with hMPV and coculture of these cells with CD4^+^ T cells, a significant decrease in the activation of the T cells was detected compared to control conditions [51,129]. This decrease may depend on the presence of the G and SH proteins of hMPV (Table 1), as viral strains deficient for these proteins induce a more robust activation of CD4^+^ T cells [77]. Remarkably, this decrease also required direct contact between DCs and T cells, as demonstrated using transwell assays [77]. Because CD4^+^ T cells are among the most significant cells that will secrete cytokines, and as indicated above, hMPV leads to a proinflammatory environment upon infection, it is relevant to evaluate this secretion [130]. Using IL-17 KO mice, it has been shown that the absence of IL-17 reduced the infiltration of neutrophils, along with the numbers of T helper 1 (Th1) (IFN-γ^+^ T cells) and Th2 (IL-4^+^ T cells) differentiated lymphocytes in the lungs of hMPV-infected mice compared to uninfected mice [130]. Nevertheless, an increase in the number of regulatory T cells (Tregs) was also detected [130]. These results suggest that IL-17 may be playing a negative role during hMPV infections, as this cytokine may be promoting the establishment of the proinflammatory environment previously described [130].

CD8^+^ T cells are critical during respiratory virus infections, as they are the primary cell type that will aid in the clearance of infected cells [52]. Lack of CD8^+^ T cells during hMPV infections resulted in more severe diseases and increased viral loads, while adoptive transfer of this cell type aids in the clearance of this virus [40,52,131]. CD8^+^ T cells (also termed cytotoxic lymphocytes (CTL)) usually secrete IFN-γ as part of the antiviral response [132]. Accordingly, IFN-γ^+^ CTLs accumulated in the lungs of hMPV-infected mice, and the levels of soluble IFN-γ were also increased in the lungs compared to uninfected mice [132]. Because an adequate CD8^+^ T-cell response is key for the clearance of hMPV, identifying and characterizing potential antigenic targets on this virus is crucial. Two epitopes from hMPV proteins (N^307-315^ and M2-1^81-89^) were shown to elicit a protective CTL response in mice (Table 1) [73]. These two epitopes were chosen out of 12 candidates, obtained through three different predictive algorithms [73]. Adoptive transfer of CTLs specific against these epitopes protected RAG^−/−^ mice from an initial hMPV infection. However, subsequent infections induced a diversification on the repertoire of CTLs of these RAG^−/−^ mice, which led to a decrease in the protective response initially obtained [73].

The response of both CD4^+^ and CD8^+^ T cells must be finely modulated to avoid negative effects on the organism [126,133,134]. Several host molecules have been described to be in charge of this modulation. Programmed death 1 (PD-1) (CD279) is an inhibitory molecule expressed on the surfaces of several hematopoietic cells and during thymic development, recognized as part of the costimulatory signals [133,134]. Monocytes, DCs, T cells, B cells, and NKT cells have been reported to express PD-1, all cell types with significant roles during hMPV infections [133,135,136]. PD-1 expression is increased in different contexts, such as viral infections or cancer [134,137,138]. Particularly for T and B lymphocytes, PD-1 is upregulated upon TCR and B cell receptor (BCR) engagement [133,139]. This receptor can deliver inhibitory signals to T cells, inducing tolerance and anergy in these cells (Figure 2C) [133]. Engagement of PD-1 with the PD-L1 molecule expressed by the antigen-presenting cell (APC) leads to delivering these signals to T cells, blocking the PI3K/Akt pathway [133]. PD-1 is crucial to modulating an adequate immune response against hMPV [134]. During hMPV infections, PD-1 is upregulated in CD8^+^ cytotoxic T lymphocytes upon TCR engagement, inducing a decrease in their activation and their capacity to secrete cytokines, reaching functionality levels below 10% [134]. Remarkably, blockage or ablation of PD-1 (through the administration of anti-PD-L1 antibodies and the use of PD-1^−/−^ mice, respectively) prevented the impairment of the activation of CD8^+^ T cells, leading to enhanced protective responses in subsequent infections with hMPV [134]. Even more, the restriction of PD-L1 by using a PI3Kδ inhibitor (IC87114) induces the clearance of the virus [140].

Besides PD-1, several other inhibitory components have been described to modulate the immune response; such are the cases of T-cell immunoglobulin and mucin-domain-containing 3 (TIM3), LAG3 (CD223), and 2B4 (CD244) (Figure 2C) [134,141,142,143,144,145]. TIM3 is a surface receptor expressed on IFN-γ^+^ T cells, along with Tregs, macrophages, and DCs, working as a coinhibitory molecule [141]. The primary role described for TIM3 upon engagement is the inhibition of the polarization of T cells towards a Th1 profile [141]. The endogenous ligand for this receptor, which will induce this inhibitory effect, is galectin-9 [134,141]. Galectin-9 is a C-type lectin highly expressed on tumor cells [142]. During primary and secondary infections of mice with hMPV, TIM3 is upregulated in T cells, aiding in the impairment of CD8^+^ T cells [134]. LAG3 is an inhibitory surface receptor, usually expressed spatially close to the CD4 receptor on T cells [143,144]. LAG3 is also expressed by CD8^+^ T cells, Tregs, NK cells, and DCs [143]. The endogenous ligand of this receptor is the MHC-II molecule, which is expressed mostly by APCs, such as DCs, macrophages, and B cells [143]. Moreover, the affinity of LAG3 for the MHC-II molecule is higher than the affinity of CD4 for the MHC-II molecule [143,144]. LAG3 has been implicated in T-cell activation and cytokine secretion [134]. During primary and secondary infections with hMPV, the expression levels of LAG3 were upregulated, as also seen for the receptor TIM3 [134]. 2B4 is another inhibitory receptor expressed on the surface of T cells, NK cells, and DCs [145]. The endogenous ligand for this receptor is CD48, expressed by most lymphocytes and other innate immune cells [145]. As seen for TIM3 and LAG3, expression of 2B4 is significantly increased in CD8^+^ T cells upon infection with hMPV [134]. Although TIM3, LAG3, and 2B4 are upregulated during hMPV infections, PD-1 remains the dominant inhibitory receptor, because blockage of any or all these receptors but PD-1 did not induce significant changes in the impairment of CD8^+^ T cells during hMPV infections [134].

Because the IFN response is one of the most relevant during viral infections, it is relevant to address the role of this cytokine during the adaptive immune response [48,126]. Infection of mice deficient for IFNAR (IFNAR^−/−^) with hMPV did not show differences in the number of total lymphocytes at day 10 p.i. [48]. Therefore, IFNAR signaling does not seem to be required for the recruitment of T lymphocytes. However, epitope-specific CD8^+^ T cells (recognizing a small fragment of the F protein of hMPV) were reduced in IFNAR^−/−^ mice compared to Wild-type (WT) mice [48]. These epitope-specific CD8^+^ T cells were also impaired in their capacity to secrete IFN-γ compared to WT mice [48]. No changes were detected in the levels of PD-1 on CD8^+^ T cells from IFNAR^−/−^ and WT mice. However, lack of IFNAR signaling resulted in a decreased expression of PD-L1 (the ligand for the inhibitory receptor PD-1) in AECs, DCs, and interstitial macrophages upon infection with hMPV, at day 5 p.i. [48]. IFNAR^−/−-^ mice exhibited increased TIM3^+^ F-protein epitope-specific CD8^+^ T cells compared to WT mice. Therefore, the inhibitory receptor TIM3 could be responsible for the impaired CD8^+^ T cell response elicited upon hMPV infection [48]. The IFN-γ response is impaired during hMPV infections, as seen in vitro and in vivo for both humans and mice [71,146]. Further studies are required to soundly identify the mechanisms that hMPV uses to impair the response elicited by T cells.

B lymphocytes and their subsequent maturation into plasma cells are responsible for the secretion of antibodies and the mounting of a proper humoral response, which is one of the essential lines of defense of the immune response [126,147]. Because antibodies are the main protagonist of the humoral response, they cannot be overlooked during infections. Commonly, inefficient neutralizing antibodies against hMPV are described upon infection in either humans or mice [40,148,149]. Therefore, the natural humoral response alone is not enough to protect against subsequent infections. Studies using IFNAR^−/−^ mice showed no differences in CD19^+^ cells (a surface marker commonly associated with B cells) at day 10 p.i. with hMPV compared to WT mice [48]. Therefore, and as indicated above, IFNAR signaling does not seem to be required to recruit B lymphocytes. However, IFNAR^−/−^ mice exhibited increased titers of neutralizing antibodies upon infection with hMPV compared to WT mice [48]. This enhanced humoral response did not change their susceptibility to secondary infections [48]. hMPV significantly modulates the response elicited by B lymphocytes, and further studies are required to comprehend how to counter this modulation soundly. A possible way to understand how this modulation works is to evaluate the immune response triggered upon an hMPV infection in the germinal centers. This understanding could be achieved through assays that lead to understanding whether the viral proteins or soluble factors induced by the infection are the ones modulating the activation and differentiation of B cells into more complex cell types such as plasma and memory cells. These studies could also be essential to understand the factors involved in the functionality of the antibodies secreted during a natural infection. Some molecular targets to evaluate this could be the Fcγ receptors or the ephrin family, where ephrin B1—present in the germinal center—can interact with different variants of the type B erythropoietin-producing hepatocellular (EPH) receptors (part of the Eph tyrosine kinases receptor), such as EPH type B (EPHB) expressed in the follicular helper T (Tfh) cells [150,151,152,153]. The EPHB receptor has been described to induce the maturation of the B cells in the germinal center into memory or plasma cells, and it also modulates the interactions between different adhesion molecules, such as ICAM-1-LFA-1, CD40-CD40L, and SLAM-SAP, among others, which are essential for proper cellular communication [150,151,152,153].

As indicated above, the role of the adaptive immune response in the modulation of the response against hMPV is crucial. Accordingly, it must be further studied since the literature lacks a significant number of reports addressing the molecular mechanisms underlying the immune response elicited against this virus. The use of various tools such as small interfering RNA (siRNA), agonists, and antagonists, the generation of mutant viral strains, the use of transcriptomics and proteomics, and different animal models are fundamental for understanding the questions still unknown to us.

## 4. Conclusions

hMPV is one of the leading viruses responsible for respiratory tract infections. This virus infects mostly cells at the lower respiratory tract (LRT), and the symptoms related to the disease may range from fever, coughing, wheezing, and bronchiolitis to the requirement of mechanical ventilation, encephalitis, and febrile seizures in the most severe cases. hMPV infections rise during winter, and the peak lasts until the end of spring, sometimes overlapping with other respiratory viruses such as influenza virus and hRSV. The burden associated with hMPV infections worldwide is heavy, with social and economic costs that cannot be overlooked, affecting up to 86% of infants under five years old. Therefore, the development of therapies for the control of this disease is fundamental.

Understanding how the cells and components of the immune system contribute to modulating this disease can aid the develop of treatments (such as antivirals and vaccines) against this virus. Since its first description in 2001, several articles have been published that address the molecular mechanisms that hMPV uses to infect its target cell and evade the immune response. However, there is still a considerable knowledge gap associated with this pathogen.

The innate and adaptive immune responses are significantly impaired during hMPV infections because many of the viral proteins can modulate the response elicited by the host. For instance, the P protein expressed by some hMPV strains is capable of inhibiting the production of IFN-I by impairing the sensing of viral RNA by RIG, the SH protein can inhibit the transcription of many ISGs by impairing the phosphorylation of STAT1, and the G and SH proteins may be responsible for impairing the activation of CD4^+^ T cells. Several PRRs can also recognize different PAMPs from hMPV, which promote the activation of both immune and nonimmune cells. As mentioned above, the activation of CD4^+^ T lymphocytes is impaired during hMPV infections, a phenomenon that requires direct contact between the infected DCs and the T cells, resulting in a poorly modulated adaptive immune response. The naturally induced humoral response alone cannot provide protection from hMPV infections because the antibodies generated exhibit inefficient neutralizing capacities. Vaccine prototypes aiming to enhance the secretion of neutralizing antibodies are promising candidates to induce a protective humoral immune response.

Finally, intrinsic antiviral factors with the capacity to directly impair the viral replication or fusion of hMPV with its target cells have been described. IFITM and TRIM are some of the most well-described antiviral factors to date. IFITM3 has been described to prevent in vitro infections with some subtypes of hMPV. However, TRIM56 seems to play no role in the inhibition of hMPV replication. Further studies are required to comprehend the molecular mechanisms underlying infections with hMPV soundly. A significant focus should be given to the intrinsic antiviral factors of the host, as these molecules play a vital role in the modulation of viral infections and their role in hMPV infections must be deciphered.

## Figures and Tables

**Figure 1 viruses-13-00519-f001:**
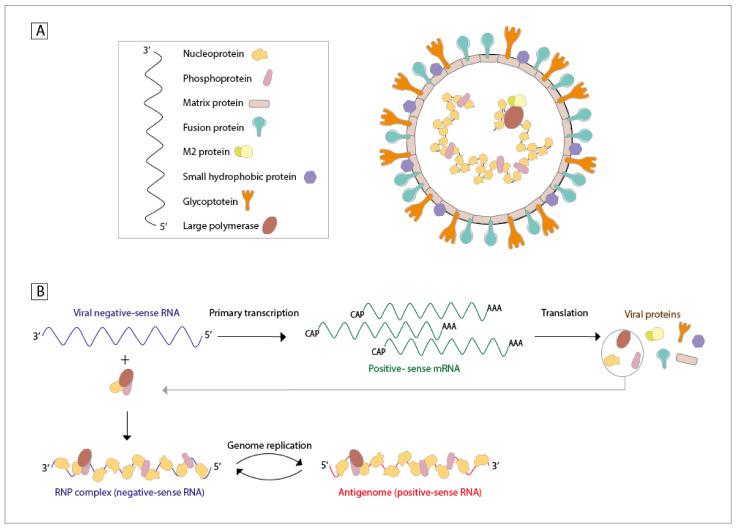
Structure, genetic material, and replication cycle of human metapneumovirus (hMPV). (**A**) hMPV is a negative-sense and single-stranded RNA virus, with nine structural proteins in its genome. These proteins are found in the genome in the following order: 3′-nucleoprotein (N), phosphoprotein (P), matrix proteins (M), fusion protein (F), M2-1/2 protein, small hydrophobic protein (SH), glycoprotein (G), and large polymerase protein (L)-5′. (**B**) The replication cycle of hMPV involves the synthesis of positive-sense mRNA, which will be translated into the indicated viral proteins. These viral proteins are essential for replicating the virus, as they will bind to the initial viral negative-sense RNA, forming the ribonucleoprotein (RNP) complex. This RNP complex will replicate the negative-sense RNA into positive-sense RNA, ultimately generating many copies of the negative-sense RNA genome for newborn viral particles.

**Figure 2 viruses-13-00519-f002:**
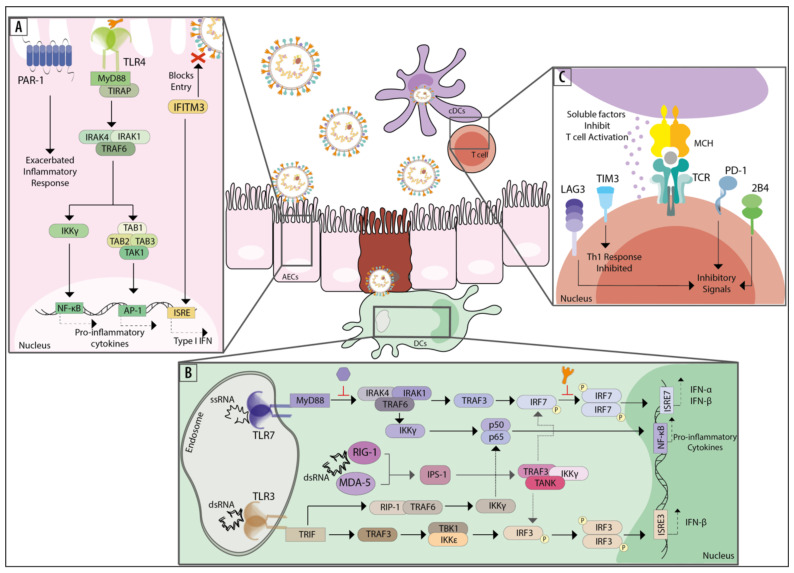
Molecular components of the host contributing to the modulation of the immune response during human metapneumovirus (hMPV) infection. (**A**) Zooming into an airway epithelial cell (AEC), the different mechanisms responsible for modulating the antiviral response against hMPV can be found. The protease-activated receptor 1 (PAR-1) exacerbates the inflammatory response. The interferon-induced transmembrane (IFITM) proteins prevent the entry of the virus into the cell while also promoting the secretion of type I IFN. Additionally, Toll-like receptor (TLR) 4 promotes the secretion of proinflammatory cytokines. (**B**) Zooming into a cell from the innate immune response, such as a dendritic cell (DC), several pattern recognition receptors (PRRs) and modulatory molecules can be found. TLR7 and TLR3 are expressed inside these DCs (within the endosomes), and these receptors recognize ssRNA and dsRNA, respectively. TLR7 induces the secretion of type I IFN (IFN-α and IFN-β) and NF-κB, while TLR3 promotes the expression of IFN-β and NF-κB. Retinoic acid-inducible gene I (RIG-I) and melanoma differentiation-associated gene 5 (MDA5) can be found in the cytosol, which induce the secretion of IFN-α and IFN-β upon recognition of dsRNA. (**C**) Zooming into a cell from the adaptive immune response, such as a T lymphocyte, the different receptors and modulatory mechanisms associated with T cell activation can be found. An impaired T-lymphocyte activation has been described when these cells are cocultured with hMPV-infected DCs. The inhibitory receptors programmed death 1 (PD-1), LAG3, and 2B4 will induce an anergic state in these T lymphocytes. T-cell immunoglobulin and mucin-domain containing-3 (TIM3) impairs the activation of the T helper 1 (Th1) signaling pathway, a response that is especially suited to fighting viral infections.

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
