# Peer review of "Host Components That Modulate the Disease Caused by hMPV"

_viruses, 2021, doi:10.3390/v13030519_

Round 1

Reviewer 1 Report

Manuscript ID: 

vaccines-1046114

Type of manuscript: Review

Title: Host components that modulate the disease caused by hMPV.

Authors: Nicolás M.S. Gálvez, Catalina A. Andrade, Gaspar A. Pacheco, Jorge A. Soto, Vicente Stranger, Thomas Rivera, Abel E. Vásquez, Alexis M. Kalergis *

General comment

The authors try to review the relationship between human metapneumovirus (hMPV) and the host immune system. Because the host immune components related to hMPV remains to explain, it will be quite important to organize what we know and do not know about the hMPV reproductive cycle. The topic may provide a new perspective on the development of hMPV treatment. However, I have several concerns over the manuscript.

It would be informative to focus only on the host components related to hMPV specific. The authors review the general host immune system rather than hMPV specific host immune system.

For specific comments, please see below.

Major comments

1) The authors should show detailed information for each hMPV proteins, including molecular biological and biochemical characteristics. It would be informative to understand each hMPV protein's role in the host immune system's involvement.  

2) To indicate the relationship between each hMPV proteins and host components, the authors should show a table, which includes each hMPV proteins, responsible host components, roles, and refs, which might be contributing to allow the readers to understand the relationship quickly.

3) In my opinion, general information to understand immune systems, such as interferons, cytokines, and immune cells, should be kept to a minimum. Most readers need the reviewed findings, just 'host components closely related to hMPV.'

Author Response

Please find attached a PDF file with a point-by-point response to the comments issued by the Reviewer.

We would like to thank the Reviewers and the Editors for their time and effort in the handling of this document, and we hope that the current revised manuscript is acceptable for publication in Viruses.

Answers to Reviewer 1

Reviewer #1: The authors try to review the relationship between human metapneumovirus (hMPV) and the host immune system. Because the host immune components related to hMPV remains to explain, it will be quite important to organize what we know and do not know about the HPV reproductive cycle. The topic may provide a new perspective on the development of hMPV treatment.

Answer: As requested by the reviewer, we have modified the manuscript to address the requests and criticisms. We believe that our article was significantly improved upon the revision and hope that the current version is acceptable for publication.

Reviewer #1: However, I have several concerns over the manuscript.It would be informative to focus only on the host components related to hMPV specific. The authors review the general host immune system rather than hMPV specific host immune system. For specific comments, please see below.

Answer: As requested by the Reviewer, we have edited the manuscript, reducing the amount of information regarding the overall immune response. Instead, we have focused on the response of the host against an hMPV infection.

Reviewer #1: The authors should show detailed information for each hMPV proteins, including molecular biological and biochemical characteristics. It would be informative to understand each hMPV proteins role in the host immune system involvement.

Answer: As requested by the Reviewer, we have included more detailed information regarding each hMPV protein.

Reviewer #1: To indicate the relationship between each hMPV proteins and host components, the authors should show a table, which includes each hMPV proteins, responsible host components, roles, and refs, which might be contributing to allow the readers to understand the relationship quickly.

Answer: As requested by the Reviewer, we have included a table indicating the relationship between the hMPV proteins and the host components described up to date(Page 5, Line 185).

Reviewer #1: In my opinion, general information to understand immune systems, such as interferons, cytokines, and immune cells, should be kept to a minimum. Most readers need the reviewed findings, just 'host components closely related to hMPV.'

Answer: As requested by the Reviewer, we have edited the sections of the manuscript addressing the interferons, cytokines, and immune cells to reduce the amount of general information presented and focus on the specific association between these responses and hMPV.

We would like to thank once again the Reviewers and the Editors for their time and effort in the handling of this document, and we hope that the current revised manuscript is acceptable for publication in Viruses.

Reviewer 2 Report

This article provides a thorough overview of the literature of host factors that are considered to modulate disease caused by hMPV. The organization and content around innate and adaptive immune responses and host determinants (intrinsic components and restriction factors) implicated are well-founded and appropriate.

There is some inconsistency throughout the text. For example, in some places, the discussion focuses exclusively on hMPV while in other places, comparisons to hRSV are made. The authors are encouraged to consider this with greater rigor and revise the manuscript accordingly.

There is, understandably, the tendency to end sections with wording along the lines of "Further studies are needed...". These types of statements would benefit from the authors providing more detailed commentary about the types of experiments that could be done to provide new information with the potential for high impact, and explain their reasoning accordingly.

Section 4. Conclusions. The way this section is written could apply to almost any pathogen. The authors are encouraged to sharpen their focus on hMPV. 

There are scattered typos (e.g., Line 352 and elsewhere: "titters" should be "titers") and occasional awkward language (e.g., Line 486: "Because the IFN response is one of the most relevant during viral infections, it is relevant to address the role..." (boldface mine) which should be easy to detect by careful proofreading and straightforward to revise.

Author Response

Please find attached a PDF file with a point-by-point response to the comments issued by the Reviewer.

We would like to thank the Reviewers and the Editors for their time and effort in the handling of this document, and we hope that the current revised manuscript is acceptable for publication in Viruses.

Reviewer 3 Report

The authors detail the host components associated with modulating hMPV infection in terms of innate and adaptive immunity. This review contains very useful information for understanding the immune response to hMPV infection.

1. Lines 127-137: The description is abrupt. Please reconsider the description location for each content.

2. Lines 132-137: There is only one hMPV serotype. hMPV is divided into two groups, A and B, according to genotype. Please describe accurately.

Author Response

We would like to thank the Reviewers for their time and effort in the handling of this document, and we hope that the current revised version is acceptable for publication in Viruses. Please see the document attached for a point-by-point response to the comments.

Round 2

Reviewer 1 Report

General comment

The authors try to review the relationship between human metapneumovirus (hMPV) and the host immune system. Although they revised the manuscript based on the reviewer’s comments, little information is available. Because the most relationship between hMPV and host immune system remains to explain yet, it’s too early to review this topic. Due to few reports compared to other respiratory viruses, they still devoted the manuscript to explaining the basic immune system.

Author Response

As requested by the Reviewer, we have further revised the current literature to include additional information in the manuscript. We have added references regarding the role of several viral proteins in the evasion of the immune response. Additionally, we have included throughout the manuscript more references relative to the host response upon infection with hMPV.